# Relevance of COVID-19 vaccine on the tourism industry: Evidence from China

**Fredrick Oteng Agyeman** [1]*, **Zhiqiang Ma**[1]*, **Mingxing Li**[1]*, **Agyemang Kwasi Sampene**[1], **Israel Adikah**[1], **Malcom Frimpong Dapaah**[2]

**1** School of Management, Jiangsu University, Zhenjiang, P.R. China, **2** School of the Environment and Safety Engineering, Jiangsu University, Zhenjiang, P.R. China

* fredrickotengagyeman2@gmail.com (FOA); mzq@ujs.edu.cn (ZM); mingxingli6@163.com (ML)

**Data Availability Statement:** The data is not owned by the authors. It was collected from the following accredited sources: https://stats.oecd.org/index.aspx?DataSetCode=TOURISM_

## Abstract

### Background

Vaccination is indeed one of the interventional strategies available to combat coronavirus disease (COVID-19). This study emphasizes the relevance of citizens' acceptance of the COVID-19 vaccine in assisting global recovery from the pandemic and aiding the tourism industries to return to normalcy. This study further presented the impact of COVID-19 on the tourism industry in China. Also, the study confirmed the past performance of tourism in China to the current tourism-related COVID-19 effects from a global perspective by employing Australia's outbound tourism data from 2008 to 2020 on top 6 destinations, including China, Indonesia, New Zealand, Thailand, the United Kingdom, and the United States.

### Methods

Jeffrey's Amazing Statistical Program (JASP) was used to analyze this study. The JASP statistical software was employed to accurately analyze the vaccines administered in China from December 15, 2020, to March 28, 2021.

### Results

The study results demonstrate an overwhelming acceptance of vaccines in China which will positively and significantly impact the globe's travel and tourism industries. Also, the study findings indicated that industries in tourism are hopeful of regaining the past losses. Further, the study results showed an enormous decline in death and new cases.

### Conclusion

Vaccine acceptance is relevant for the eradication of the COVID-19 pandemic. Therefore, neighborhood and individual-level acceptance of the vaccine will help reduce the challenges facing the tourism industries and the world. The researchers recommend that authorities should strictly check the vaccination certificates of visitors. Furthermore, hoteliers should put adequate measures to monitor all visitors who visit the various tourist destinations.

OUTBOUND and https://www.who.int/news-room/q-a-detail/coronavirus-disease-(covid-19)-vaccines. Individual researchers may access the data directly through the links provided without any restrictions or special privileges required. The data underlying the data's study can be also found on figshare through this DOI: 10.6084/m9.figshare.19799947.

**Funding:** ML National Natural Science Foundation of China grant number (No.71974082) NO, The funders had no role in study design, data collection and analysis, decision to publish.

**Competing interests:** The authors have declared that no competing interests exist.

## Introduction

The emergence of COVID-19 has posed a significant risk to the tourism industries across the globe and made the tourism industries find it very cumbersome to achieve set objectives [1]. The negative impact of COVID-19 on citizens' mental health, including psychological, emotional, and social well-being, and the tourism industry has necessitated the search for effective vaccines across the globe to curb the pandemic [2, 3]. Vaccines availability assures and relieves the citizens' psychological and emotional stress conditions [4, 5]. Notwithstanding, many countries are still battling the pandemic while others confront vaccine acceptance issues, which needs global cooperation. Resetting the COVID-19 pandemic recovery button as an opportunity to start over would be driven by sustainable tourism development principles [6–12]. The temporary shock rendered by COVID-19 to the tourism industry has challenged the transportation industries, security, and development of many economies [13–15]. The current tourism's success relies on the availability of effective COVID-19 vaccines.

Tourism is perceived as a social, cultural, and economic concept involving individuals traveling to nations or places outside of their usual environment for personal or professional reasons for not more than one consecutive year. High-quality tourism development is increasingly captured as an essential key in enhancing economic growth and alleviating poverty [16]. Before the COVID-19 pandemic, tourism in the picturesque of the world is arguably the most prominent and fastest-growing industry, accounting for about 5% of the world's gross domestic product (GDP) and 6% of employment opportunities [17, 18]. The contributions of tourism industries to GDP, foreign direct investment, and employment generation among remote, rural, and urban areas are overwhelming [19].

Peeking in on China, the COVID-19 outbreak has had a massive ramification on the country's economy in the short term. The effect on net exports was anticipated to extend from the medium to the long term [20]. It is maintained that making effective policies and delivering vaccinations is not a competitive racing to the finish line but rather a careful assessment of a fast, effective, worldwide response [21, 22]. The absence of vaccines to treat infected individuals increases the number of virus infections and mortality. At the start of the coronavirus pandemic, herd immunity was proposed as a possible treatment [22]. As of May 28, 2021, the World Health Organization (WHO) had received reports of 168,599,045 confirmed COVID-19 cases worldwide, with 3,507,477 deaths. An aggregate of 1,546,316,352 vaccine doses had been dispensed worldwide as of May 26, 2021 [23].

Presently most countries are overwhelmingly accepting vaccines to control the COVID-19 virus. The vaccine has increasingly given hope to the industries, especially the tourism industries, as a sign of revamping. In China, the importance of vaccine administration is centered on four key reasons [24, 25]:

- Once the mobility of people worldwide resumes, people who are not vaccinated will be at risk.

- Vaccination is the most effective prevention and control measure of COVID-19. Therefore, it is necessary to inoculate the vaccine to assist many people in attaining immunity and protection against the virus as soon as possible.

- The faster the vaccination, the earlier the herd immunity is formed. Therefore, vaccination is the best strategy to attain herd immunity.

- The immune barrier (herd immunity) will not be established if citizens do not vaccinate. As a result, the epidemic can reappear once the source of infection exists, affecting people's lives, jobs, and studies.

The virus's severity and mode of spread made many countries and the WHO search for an effective vaccine to control the COVID-19 pandemic [26]. WHO's involvement has led to the increasing acceptance of the vaccine doses, giving hope to tourism, manufacturing industries, humanity, and the world [27–29]. To curb the spread of coronavirus prevalence in China and reduce the outspread to the world, Governments of different countries planned to test and vaccinate their citizens to attain absolute freedom from the pandemic. For example, in China, the National Immunization Program, which serves as a joint venture between the government, citizens, and academia, implemented numerous guidelines for vaccine administration. Its ultimate aim is to help China monitor, eliminate, or eradicate virus-preventable diseases by increasing public awareness of the benefits of immunization and encouraging vaccine knowledge and use [30].

Hence, this study aims to establish the relevance of the COVID-19 vaccine and the impact of the pandemic on China's tourism. Therefore, this research seeks to: analyze the relevance of vaccine administration on tourism industries amidst the COVID-19 pandemic in China. This current research also investigates the past and current trend of China's tourism activities in relation to Australia's outbound total international tourism data from 2008–2020 on the top 6 destinations, including China, Indonesia, New Zealand, Thailand, the United Kingdom, and the United States whose impact of COVID-19 pandemic was enormous [31].

The current research makes various contributions to tourism studies and the COVID-19 debate. Although the impact of the COVID-19 vaccine doses on tourism success has been well documented, investigating the relevance of COVID-19 vaccine doses administration is fascinating. The contributions of the current study are as follows: the study highlights the relevance of the COVID-19 vaccine doses, the impact of the COVID-19 on tourism, and the JASP statistical method used for the analysis. In the researchers' view, this is the first study to investigate the relevance of the COVID-19 vaccine doses and the impact of the COVID-19 on tourism using JASP statistical analysis.

This study also contributes to the literature on the COVID-19 vaccine doses administration and the impact of the COVID-19 on tourism by assessing how tourism industries might use dynamic approaches to innovate, sustain tourism development, and succeed in the current unstable and unforeseeable conditions.

The remainder of this research is coherently organized: a review of the related literature, the methodology, and data analysis; the results and discussion of the research; and finally, the conclusion and policy implementation.

## Literature

The terrifying announcement of seasonal flu occurrences, epidemic diseases, pandemics, and catastrophic events leads to a sharp decline in the travel and tourism sector, a significant contributor to the service sector [14, 32, 33]. Pandemics negatively influence tourist behavior and psychological health [1]. Tourists usually cancel their scheduled trip arrangements out of anxiety of virus transmission, as it appears implausible to resist virus infection during trips [34–36].

Furthermore, in the absence of robust vaccines, tourism-related travel increases the risk of infection to many other travelers [37–39]. The COVID-19 pandemic and its prevalent new variant of SARS-CoV-2 and Delta variant have negatively impacted the tourism industry and other human activities due to the restriction of movement [16]. Thus, the emergence of the deadly COVID-19 disease has led to substantial economic losses, worldwide health issues, and economic catastrophes around the globe [40]. The unavailability of a precise vaccine for treatment at the early stages of the COVID-19 pandemic created an avenue for various trial

treatment methods. For instance, a mixture of hydroxychloroquine and azithromycin was adopted in France to treat COVID-19 infected patients [41].

Countries were heavily dependent on the tourism sector as a good source of revenue and employment for their citizens. As numerous gains are derived from the tourism industry, its sustainability looks agile in the pandemic periods, leading to numerous downsides in this industry [42]. Tourism, which served as one of the vibrant, dynamic, and significant employment-intensive sectors in the contemporary period [33], has been seriously hit by the COVID-19, causing unemployment and financial loss to industries, individuals, countries, and the globe at large [20].

Based on the International Monetary Fund (IMF) predictions on tourism within the after-pandemic world, tourism receipts are likely to rebound to the level of 2019 by 2023 or later [43, 44]. The initial stage of the pandemic within the year 2020 saw a gradual decline of tourists globally, counting in percentage as 65 compared with the 8% decline experienced in the global financial contagion era and 17% during the surge of the SARS pandemic in 2003 [45]. In 2020, the World Economic Outlook (WEO) predicted that the world economy would contract by 4.4%. Thus, the COVID-19 impact on countries relying on tourism would worsen. The GDP of nations that rely significantly on tourism will shrink by 12% [46, 47].

Further, the IMF release in 2020, concerning the G20 nations that engage in hospitality and travel businesses were previously contributing to employment by 10% and 9.5% of GDP in Italy and Spain, respectively. However, six-month halt inactivity may diminish the GDP by 2.5% to 3.5% in all G2O countries [46]. The decisions for the tourism recovery should aim at inclusive growth, a road map for the difficult times ahead: unleashing development and investment, boosting innovation and technology, facilitating travel, fostering resilience, and rebranding [48].

The International Labour Organization's (ILO) purpose of creating suitable job opportunities has become entangled with the impact of COVID-19-induced economic catastrophe, which has resulted in a reduction in tourism workers, most of whom are paid insufficiently or not at all [49–51]. Also, the COVID-19's global influence on job and employment possibilities, particularly for workers in the informal economy, had been greatly affected. Due to the near-collapse of the foreign tourism industry, many of these workers are currently refused permission to work [52]. Many employees worldwide operate in the gig economy, where workers' social security and job rights are restricted or non-existent. Job security problem has raised issues about migrant workers' vulnerability regarding contracting the viral disease and their job rights [53]. At the best of times, tourism activities are fragile [54] and have further been worsened by the emergence of the COVID-19 pandemic than the other sectors. It remains to be seen whether the lost personnel in the tourism sector will generally return to serve tourists following the outbreak [54].

The prohibitions imposed during the surge of COVID-19 heightened the contraction of global tourism industries, necessitating effective and efficient policies to assist industrialists while investigating the causes and appropriate vaccines to be used. The turbulence and the trial-and-error period are almost over. An effective vaccine has been produced in China and other countries. This declaration by the [55] has further heightened the confidence of the tourism sector to assume their duties as more vaccines are currently available for use. The first extensive vaccination campaign began in early December 2020, and by February 15, 2021, about 175.3 million vaccine doses were inoculated [55].

The afore literary works analyzed in this study reveal that the COVID-19 outbreak has wreaked havoc on the worldwide economy, with drastic repercussions for all individuals and their psychological conditions. The epidemic has profited from internationalization and inherent interconnection, changing positive global interconnections into a worldwide economic

shock. The shocks from COVID-19 due to international connections have unjustifiably influenced the most vulnerable countries. The studies reviewed further show the interconnection inherent in the global Sustainable Development Goals (SDG). Hence, it jeopardizes worldwide efforts to attain SDG [56]. All these key reasons represent the shambles COVID-19 is grappling with tourism and necessitate accurate identification and scientific modeling to emphasize the relevance of the vaccine on the pandemic and determine the path of tourism.

## Methodology and data analysis

This study employed Jeffrey's Amazing Statistical Program (JASP) linear regression model for analysis. JASP Linear Regression is a statistical tool for analyzing the relationship between predictor and outcome variables. Therefore, the overwhelming acceptance of JASP for scholarly research is highly recommended [57–59]. The Linear regression analysis of JASP was used to assess whether one or more of the predictor variables explain the dependent (criterion) variable in this study. Linear regression is a widely recognized scientific method that is reliable for analyzing and predicting data. The properties of linear regressions facilitate easy understanding and replicability. Further, the linear regression model has a significant explanatory predictive power. Linear regression models help to explain study data with a minimum number of parameters or predictor variables.

Also, inferences would be drawn from the JASP statistical analysis to determine the relevance of the COVID-19 vaccine on curbing the death and new cases arising in China and its corresponding effect on tourism industries. The researchers used COVID-19 vaccine doses, newly confirmed cases, and new death data from the World Health Organization [23], spanning December 15, 2020, to March 28, 2021, to investigate the trend of COVID-19 vaccine administration and its impact on individuals and tourism industries. Further, Australia's outbound data from 2008 to 2020 [31] was used to determine China's tourism industry's past and current performance. This study used vaccine doses as the dependent variable and the death and recorded new cases as independent variables. This relationship is indicated in Eq 1:

$$y = c + b_1x_1 + b_2x_2 + \cdots + b_nx_2n \tag{1}$$

Where $y$ is the dependent variable (vaccine), $c$ is the constant, $b_1$, $b_2$ is the beta, $x_1$, $x_2$, $x_n$, are the independent variables. Eq 1 is designed to align with the JASP software automatic computations for the variables. Some salient analytical techniques helpful for analysis of this study in the JASP software for modeling are T-test, ANOVA, F-test, Correlation coefficient, Collinearity Diagnostics, and Residual Statistics. These tools are used to determine and establish accurate quantitative results on the adopted variables. The model summary and coefficient of determination thus the R-Squared ($R^2$) were also estimated.

## Results and discussion

### Analysis of COVID-19 vaccine doses impact on death and new cases in China

Fig 1 shows the vaccine doses' pattern and the impact of the vaccine on the death and new cases in China [23]. The dominance of the vaccine demonstrates the hope for recovery as many people are now accepting the vaccine. Acceptance of the vaccine is the pathway for tourism businesses to return to normalcy. This study conducted a statistical analysis to justify these trends. The low recorded death cases and newly recorded cases in Fig 1 demonstrate the positive impact of the vaccine dose on the populace and the total decline of death cases.

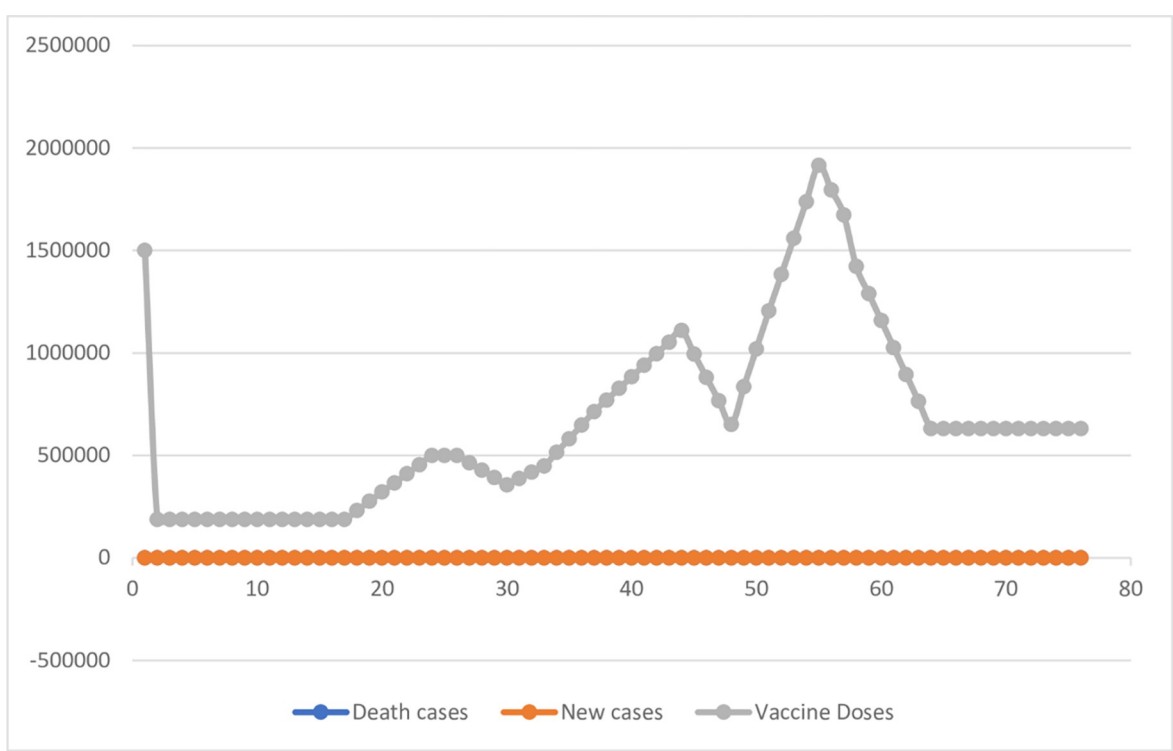

**Fig 1. Pattern of vaccine doses, death and new cases in China.** Source: [23] Authors estimation.

## COVID-19 impact on domestic tourism in China

China has emerged as a significant player in the global outbound tourist market. The gross merchandise value of China's online outbound travel industry increased by 21.7% in 2018, higher than the value obtained within the COVID-19 intense period [60]. The growth rate was expected to remain consistent through 2020, at roughly 20% [60]. The statistic shows that the proportion of outbound tourism expenditure in China's GDP to the United States increased from 0.8% in 2008 to roughly 1.8% in the 3rd quarter of 2019 [61]. However, the increasing trend declined in the 4th quarter of 2019 when the pandemic began. Thus, the soaring of China's outbound tourism experienced in the 1st to 3rd quarter of 2019 dwindled due to the COVID-19 pandemic.

Thus, besides the enormous contributions of the tourism industries in China over the past years, COVID-19 has rendered many industries incapacitated and bankrupt. Fig 2 illustrates the percentage of domestic tourists and travel revenue losses in China due to the COVID-19 pandemic in the first half of 2020 and the projected loss rate. It is expected that the vaccine's inception will lead to a decline in the spread and death cases of the COVID-19, but the pandemic's effects on tourism revenue could be severe if the spread continues. Thus experts predicted that domestic tourist revenue in China might drop by 62% in the first half of 2020, followed by a 43% drop for the entire year [62].

**The COVID-19 impact on airline travel.** According to airline industry insiders, the travel limitations brought by the emergence of COVID-19 resulted in a worldwide drop in demand, putting the loss-making carrier on the verge of bankruptcy [64]. In addition, some major airlines' implemented salary cuts ranging from 10% to 100% and unpaid leave in some destinations. These happened when traveling ban policies implemented by some countries came into existence [65].

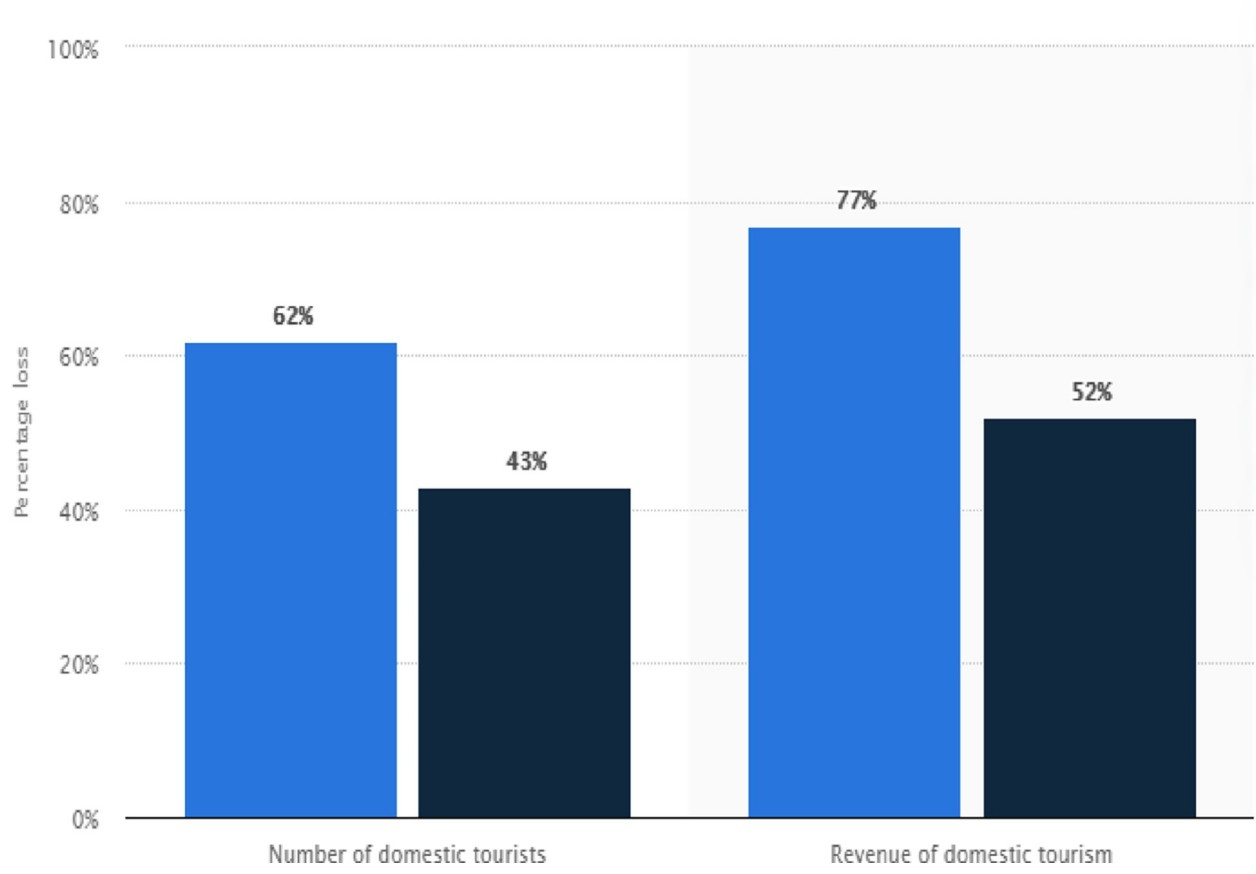

**Fig 2. COVID-19 impact on domestic tourists and revenue in China. Source:** [63].

Furthermore, several nations imposed stricter regulations on international airlines, declaring that companies can only sell one flight per week [66]. Moreover, these flights must be at least 75% filled; thus, the fear of new COVID-19 infections being "imported" into the countries prompted these regulations [66]. As a result, all major airlines operating within countries affected by the pandemic experienced a significant loss of sales and profit, putting them at risk of bankruptcy. Consequently, some airline companies asked the respective governments to intervene to support the airline industry.

**Impact of COVID-19 on hotel industries in China.** For the hoteliers in China, 2020 proved to be a grim year. Most Chinese New Year travel plans were cancelled due to the pandemic; this affected the tourism industry, which yields more than half a trillion yuan per year. Further, hotels in China experienced a nearly 80% decline in occupancy rate in February 2020 due to national lockdowns and travel restrictions. According to [67], a database on sampled hotels in China comprising Beijing, Shanghai, and Sanya, as shown in Fig 3, portrays that COVID19 has wreaked havoc on these hotels, especially between January and March 2020. Luxury and independent hotels have been struck the most among the hotel with standards.

Fig 3 further demonstrates the severity of the impact of COVID-19, which severely harmed the hotel business in China. In the first half of 2020, the average occupancy rate of luxury hotels in Beijing declined by 71.4% compared to the previous year, while revenue per available room (RevPAR) plummeted by 74.8%. However, high-end hotels in Sanya, a favorite tropical holiday

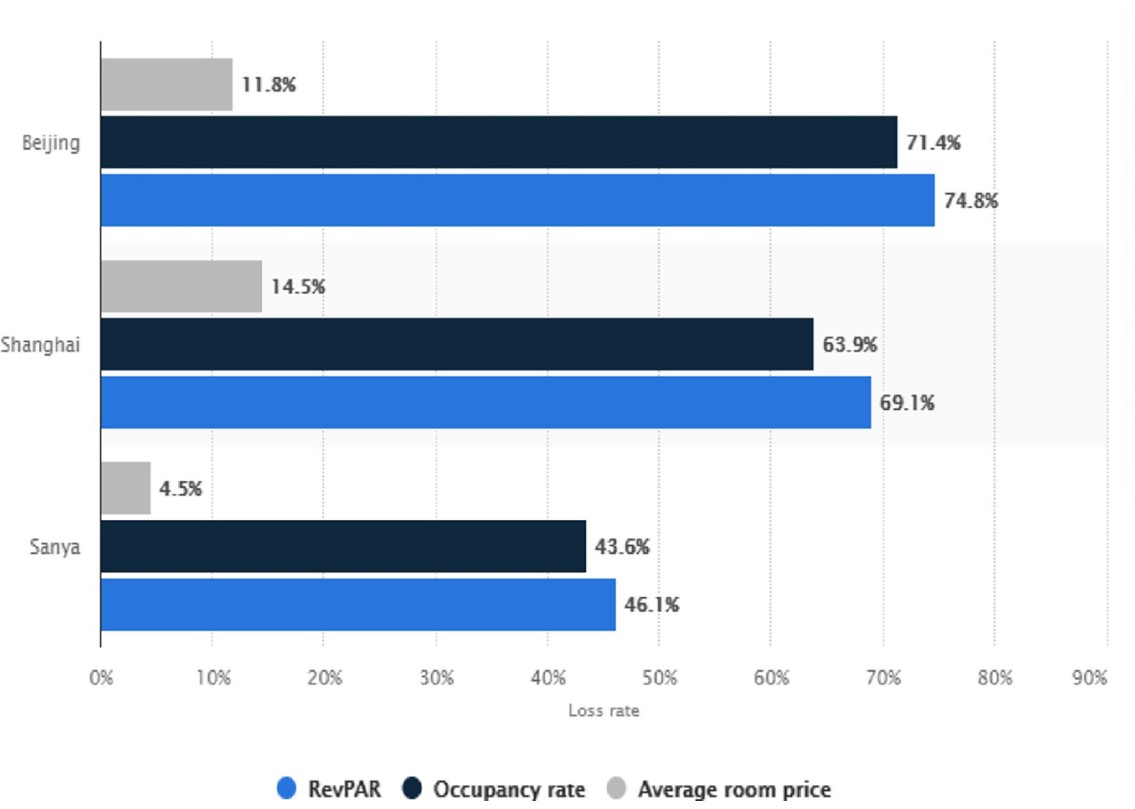

**Fig 3. The decline in occupancy rate, RevPAR, and average room price of hotel industries.** Source: [67].

area among high-end hotels, were not much affected. It witnessed only a 4.5% decline in the average room price in the same period [67].

Notwithstanding, China's hotel industry appears to be recovering in the long run, but it is not without its challenges [68]. The vaccine's arrival is the hope for the employees to bounce to work and the hotel industries to regain their losses [68]. Interestingly, the world situation influences tourism industries' recovery when people hesitate to accept the vaccine.

**Recovery and revival of the tourism industries in China amid COVID-19 pandemic era.** Though some countries are still dealing with a growing percentage of coronavirus infections, China seems to be gradually recovering from the pandemic and hopes to see tourism revamp and steadily recover. According to Smith Travel Research [69] statistics, China's daily hotel occupancy in Beijing increased to 31.8% on March 28, up from a low of 71.4% in the first week of February 2020. In addition, opening rates in important markets across the region have been significant. On March 1, Shanghai's hotel occupancy rate was as low as 11%, but by March 28, it had risen to 28.6%. The highest absolute occupancy percentages were recorded in Xi'an, 35.9% on March 28, and Chengdu, among the essential STR-designated markets for Mainland China, registered 35.6% on March 28 [69]. In Wuhan, the occupancy trend took a pretty different route. The city's occupancy rate dropped to 7.5% on Jan 23 and spiked to 72.7% on March 7, but it subsequently trended downward to 62.4% on Mar 28. Amid the fluctuations experienced, there is an encouraging sign for the tourism businesses revival. Thus, about 87% of hotels in Mainland China reopened after many days of closure in the preceding months [69]. **S1 Fig in S1 File** demonstrates Mainland China's hotel industry's early signs of performance and recovery.

It is established that revenue decreases of tourism and industries have repercussions for jobs and business models, affecting United Nations Development Programme's (UNDP) and China's ambition for the 2020 SDGs. Due to the COVID-19 shocks, many businesses were obliged to cut expenses by decreasing employee hours, slashing compensation, and, in some cases, laying off people [70]. As the pathway to recovery and revival of the tourism industries, companies must start to adapt their business strategies to get back on their feet, relying more on technology transformation [20].

## Top 6 destinations of Australia's outbound tourism–eyeing China in the COVID-19 pandemic era

This study further proved the impact of the pandemic on China's tourism by switching from China's outbound tourism to Australia's outbound tourism relating to the top 6 destinations by eyeing China's international movement. Outbound tourism refers to the actions of citizens of a particular country who travel to and stay in locations outside of their home country and unfamiliar surroundings for less than a year for recreation, business, or other purposes. Australia and China are Asia-Pacific Economic Cooperation (APEC) members, the East Asia Summit, and the G20, among other economic, cultural, and political groups. China is Australia's top trading partner and has invested in the country's industries. China's economic growth has made it a fundamental trading and investment partner for Australia. China's interest in maintaining its role in Australia's regional and global affairs has grown due to its performance. **S1 Table in S1 File** and Fig 4 showed that China, Australia, Indonesia, New Zealand, Thailand, the United Kingdom, and the United States experienced a decline in tourism revenues and engagement due to the COVID-19 pandemic within 2020 [31].

Fig 4 and S1 Table in S1 File further portray that all the top 6 destinations in Australia's outbound tourism were heavily affected by the COVID-19 pandemic. Based on the COVID-19 impact on the top six countries, the authors further conducted a statistical analysis to establish China's impact on tourism and the potency of the vaccine to ensure recovery in this study by employing the JASP Linear modeling statistical tool to analyze the effect of the vaccine acceptance doses in China.

## JASP linear regression model statistical results on vaccine doses, new death and new cases of infection in China

**Model summary, parameters, and curve estimates.** The JASP software applied to the data gathered reveals accurate point plotting. Model summary of the dependent variable, parameter estimates (ANOVA), coefficient, a summary of descriptive statistics of dependent and independent variables, Collinearity diagnostics, and residual statistics. Salient among the displayed parameters such as $F$ and $t$ distributions, degree of freedom (D.F.), significance, unstandardized and standardized statistics are estimated.

Table 1 denotes the descriptive statistics of the variables selected for the study. The variables (the new deaths, new cases, and vaccine doses) have a total observation (n = 435); there was no missing variable. The mean values for new deaths, new cases, and vaccine doses are 11.13, 233.87, and 117673.01, respectively. The means are positioned to the right of the medians for each dataset. Based on the p-value of Shapiro-Wilk, all the results were significant and less than 0.001. The Shapiro-Wilk test further indicates that the study data is normally distributed. Thus, the three variables indicate positive skewness. The outcome for the kurtosis concerning the variables is leptokurtic. The standard errors of the kurtosis and skewness demonstrate that the study data is normally distributed. This outcome signifies that the mean values are accurate for this analysis.

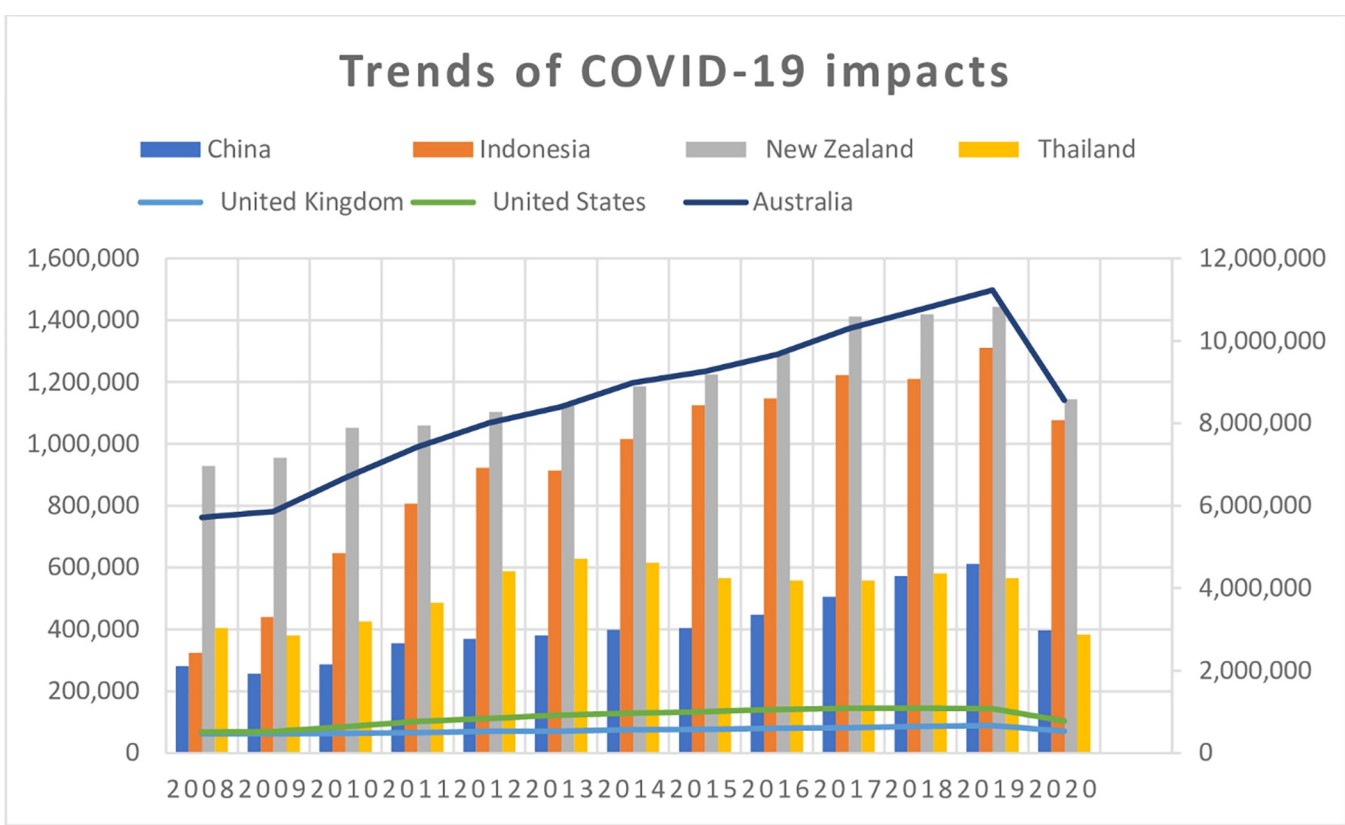

**Fig 4. Impact of COVID-19 on top 6 destinations of Australia's outbound tourism.** Source: [31] modified by the researchers.

Table 2 indicates the model summary for the dependent variable based on the Durbin-Watson independent observations. The Durbin-Watson test is expected to be closer to 2 and not more than 3. The Autocorrelation showed a value of 0.94 for both null and alternative

**Table 1. Descriptive statistics for the new death, new cases and vaccine doses (n = 435).**

| Measurement | new_deaths | new_cases | Vaccine Doses |
|---|---|---|---|
| Valid | 435 | 435 | 435 |
| Missing | 0 | 0 | 0 |
| Mean | 11.13 | 233.87 | 117673.01 |
| Median | 0.00 | 38.00 | 0.00 |
| Std. Deviation | 67.17 | 960.50 | 314717.16 |
| Skewness | 16.28 | 10.26 | 3.18 |
| Std. Error of Skewness | 0.12 | 0.12 | 0.12 |
| Kurtosis | 304.84 | 139.92 | 10.64 |
| Std. Error of Kurtosis | 0.23 | 0.23 | 0.23 |
| Shapiro-Wilk | 0.13 | 0.21 | 0.44 |
| P-value of Shapiro-Wilk | < .001 | < .001 | < .001 |
| Minimum | 0.00 | −1.00 | 0.00 |
| Maximum | 1290.00 | 15136.00 | 1.92e +6 |

**Source:** Authors estimation (JASP Software)

**Table 2. Model summary–vaccine doses.**

| Model | R | $R^2$ | Adjusted $R^2$ | RMSE | Durbin-Watson | |
|---|---|---|---|---|---|---|
| | | | | | Autocorrelation Statistic | p |
| $H_0$ | 0.00 | 0.00 | 0.00 | 314717.16 | 0.94  0.11 | < .001 |
| $H_1$ | 0.07 | 0.01 | 0.00 | 314614.85 | 0.94  0.12 | < .001 |

**Source:** Authors estimation (JASP Software)

hypotheses. It has a p-value of 0.001, which is less than 5%. Based on the model, *R* correlated at 0.00 and 0.07, demonstrating that the assumption of Durbin Watson has been satisfied. The Coefficient of determination was estimated as $R^2 = 0.01$, denoting that the predictive variables account for 1% of the differences in the dependent variable. Thus, the two predictive variables explain 1% of the impact of the dependent variable on the COVID-19 pandemic. It demonstrates that the vaccine doses had a positive and significant impact on curbing the COVID-19 daily cases and death cases. Thus, the positive impact of the vaccine denotes that the tourism industry would bounce back to their normal activities as citizens continue to accept the vaccine inoculation.

Table 3 shows the analysis of variance (ANOVA) for the regression. It shows the Sum of Squares for the regression 2.26e +11, the residual of 4.28e +13, and the degrees of freedom are 2 and 432, respectively. It also indicated that the variables have an insignificant interaction with a p-value of 0.32, greater than 0.05. The model describes that the predictors could not explain the vaccine doses better when the mean is not considered. Hence, we reject the alternative hypothesis of the predictors and accept the null hypothesis. The insignificant p-value of 0.32 suggests that the square residuals' variations did not account for significant heteroscedasticity [71, 72]. Thus, the model results are normally distributed. The intercept is omitted as no meaningful information could be obtained.

Table 4 shows the correlation coefficients measurement, which per assumption lies between -1 to 1. The standardized coefficient for the new deaths and new cases are -0.04 and -0.05, respectively. The scores show intercept ($H_1$) unstandardized value as 123498.62, and the ($H_0$) intercept unstandardized value as 117673.01. The mean of the variables measured indicated a positive value of 117673.01. However, based on the model assumptions, the unstandardized coefficients are used, indicating -193.76 for the death cases and -15.69 for new cases. Thus, there is a perfect negative relationship between new deaths and new cases reported as predictors of the vaccine doses. The negative interaction of the predictors demonstrates that a 1% increase in the quantities of the vaccines leads to a decrease in the new death and newly recorded cases by 193.76 and 15.69 points, as expressed in Eq 2.

$$y = 117673.01 + 123498.62x_1 - 193.76x_2 - 15.69x_3 \qquad (2)$$

**Table 3. ANOVA results for new death, new cases, and vaccine doses.**

| Model | | Sum of Squares | df | Mean Square | F | p |
|---|---|---|---|---|---|---|
| $H_1$ | Regression | 2.26e +11 | 2 | 1.13e +11 | 1.14 | 0.32 |
| | Residual | 4.28e +13 | 432 | 9.90e +10 | | |
| | Total | 4.30e +13 | 434 | | | |

Note: The intercept is omitted as no meaningful information could be obtained.
**Source:** Authors estimation (JASP Software)

**Table 4. Correlation coefficients result for new death, new cases, and vaccine doses.**

| Model | | Unstandardized | Standard Error | Standardized | t | p | Collinearity statistics | |
|---|---|---|---|---|---|---|---|---|
| | | | | | | | Tolerance | VIF |
| H₀ | (intercept) | 117673.01 | 15089.53 | | 7.80 | < .001 | | |
| H₁ | (intercept) | 123498.62 | 15590.38 | | 7.92 | < .001 | | |
| | new_deaths | −193.76 | 237.01 | −0.04 | −0.82 | 0.41 | 0.90 | 1.11 |
| | new_cases | −15.69 | 16.57 | −0.05 | −0.95 | 0.34 | 0.90 | 1.11 |

**Source:** Authors estimation (JASP Software)

Also, all the p-values for the slope and intercept are significant and are less than 0.001. However, the p-values for the predictors displayed insignificant values, 0.41 and 0.34 for new deaths and new cases. These results indicate that vaccine doses administered positively affect the new deaths and new cases of the COVID-19 virus infection. The regression results suggest that the model selected for the data is accurate for the analysis.

Table 4 further describes the collinearity statistics of the variables investigated. In assessing whether a variable has multicollinearity, the tolerance threshold should be more than or equal to 0.01, and the variance inflation factor (VIF) values should be substantially below 10. The variance proportion of the predictors illustrates no problem with the multicollinearity of the variables. The results indicate no multicollinearity as the predictors could not explain the dependent variable. This result demonstrates that the predictors are not too strongly correlated. As a result, all variables were within the tolerance range and the VIF. The result indicates that the explanatory variables are not multicollinear.

From the test results, it can be explained that: vaccine doses impact new deaths and newly recorded cases. The T-test results of vaccine doses representing 7.92 with the p-value of 0.000 are significant at 0.05 and a regression coefficient of 123498.62. Thus, vaccine doses positively affect new deaths and new cases. Also, the T-test results for new deaths indicated a value of -0.82 and an insignificant p-value of 0.41. They suggest that the p-value is above 0.05 and has a regression coefficient of -193.76. The result portrays that COVID-19 vaccine doses positively affect new death cases. Further, the T-test results of the newly recorded cases indicated -0.95 with an insignificant p-value of 0.34, which is greater than 0.05, and a regression coefficient of -15.69. Thus, COVID-19 vaccine doses positively affect new daily cases.

Table 5 indicates the collinearity diagnostics test for new death, new cases, and the vaccine doses. The variance proportion of the predictors illustrates no problem concerning the multicollinearity of the variables. Thus, the results indicate no multicollinearity as the predictors could not explain the dependent variables. This result demonstrates that the predictors are not too strongly correlated. The result portrays that collinearity diagnostics confirm no serious

**Table 5. Collinearity diagnostics test for new death, new cases and vaccine doses.**

| Model | Dimension | Eigenvalue | Condition Index | Variance proportions | | |
|---|---|---|---|---|---|---|
| | | | | (Intercept) | new_deaths | new_cases |
| H₁ | 1 | 1.50 | 1.00 | 0.15 | 0.21 | 0.23 |
| | 2 | 0.85 | 1.33 | 0.78 | 0.26 | 0.03 |
| | 3 | 0.65 | 1.52 | 0.06 | 0.53 | 0.74 |

Note: The intercept is omitted as no meaningful information could be obtained.
**Source:** Authors estimation (JASP Software)

**Table 6. Residual statistics test for new death, new cases, and vaccine doses.**

|  | Minimum | Maximum | Mean | SD | N |
|---|---|---|---|---|---|
| Predicted Value | −162810.79 | 123514.31 | 117673.01 | 22815.08 | 435 |
| Residual | −123514.31 | 1.79e +6 | 6.74 −11 | 313889.09 | 435 |
| Std. Predicted Value | −12.29 | 0.26 | 0.00 | 1.00 | 435 |
| Std. Residual | −0.20 | 2.00 | 0.00 | 1.00 | 435 |

**Source:** Authors estimation (JASP Software)

multicollinearity issues in the study results. All the eigenvalues are greater than 0, demonstrating that the predictors are not highly correlated and that minimal variations in the study data may not lead to more significant changes in the estimation of the coefficients. These fascinating results depict that the COVID-19 vaccine effectively curbs the incidence of new infection cases and new death. Thus, there is an assurance that the vaccine's effectiveness will help the tourism industries bounce back to their normal operations and regain all their past losses if citizens continue to accept the vaccine.

Table 6 demonstrates the residual statistics test results for the variables used for this study's analysis. The minimum values for the standardized predicted and standardized residual is (-12.29) and (-0.20), while the maximum values for the standardized predicted and standardized residual indicated (0.26) and (2.0), respectively. The result demonstrates that the standardized variables (the residuals and predicted values) have a mean of 0.00 and a standard deviation of 1. Thus, the residuals are normally distributed because they fall between -2 and 2. Further, the results depict no multivariate outliers [71, 72]. This exciting result is further represented by the residual versus covariate plot as indicated in S2 Fig in S1 File. S2 Fig in S1 File demonstrates the residual trend versus the covariate modeling analysis of the study variables. Thus, S2 Fig in S1 File depicts a significant fit between the residual and standardized residual values and supports the model approximation to the actual data performance and are considered normally distributed and accurate. These interesting findings demonstrate that the COVID-19 vaccine is potent in controlling the COVID-19 new infection cases and new death. Therefore, the vaccine's potency will assist the tourist centres in starting operation fully and may recoup all their previous losses made during the pandemic surge.

Additionally, S3 and S4 Figs in S1 File indicate the validity of the model based on the quantile-quantile (Q-Q) plot standardized residuals for normality distribution and standardized residual histogram for the study variables demonstrating the strong fit of the model employed in this study accordingly. The histogram is roughly bell-shaped; therefore, it is an indication that it is reasonable to assume that the model is normally distributed. The Q-Q normal probability distribution plot pattern is straight; hence, it provides evidence that it is good to assume that the model has a normal distribution. The Q.Q. plot demonstrates that the statistical distribution can approximate the study data.

## Graphical data analysis

**Distribution plots.** Fig 5 indicates that the trend of new deaths has reduced drastically. The results portray the efficacy of the vaccine doses administered.

Fig 6 demonstrates that the trend of new cases recorded has significantly reduced. Thus, the results portray the vaccine doses' effectiveness to eradicate the COVID-19 pandemic. This exciting result will boost the interest and confidence of the tourism businesses.

Fig 7 indicates that the trend of new cases and recent death recorded has dwindled significantly. The results show the vaccine doses' efficacy in curbing COVID-19 virus infections.

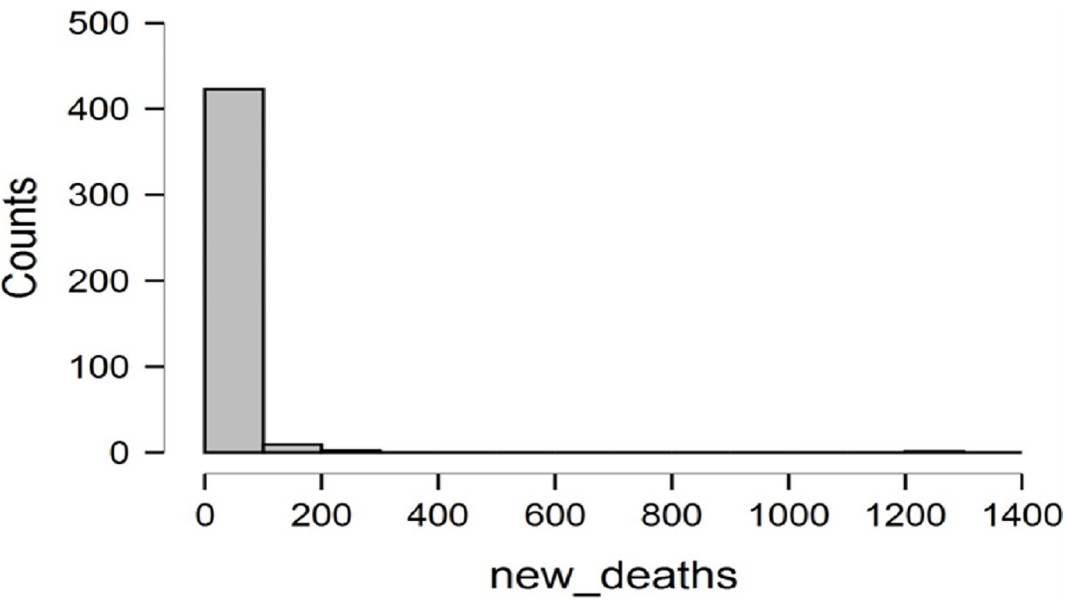

**Fig 5. Trends of new deaths. Source:** Authors estimation (JASP Software).

This fascinating result will boost the confidence and interest of the businesses and individuals in carrying out their new normal activities.

**Scatter plots.** Fig 8 demonstrates the relationship between new cases to new deaths against the vaccine doses. It demonstrates that the connection between new cases and the new deaths are dispersed. Thus, new cases of infection will cause an increase in fatalities. The result also portrayed that reported deaths and new cases will decline as vaccine doses are continuously accepted and distributed. These results further strengthen the hope and eliminate the anxiety of many tourism industries, manufacturing companies, and individuals to revamp all their losses during the surge of the COVID-19 pandemic.

Fig 9 depicts the scatter plot of new cases against the vaccine doses. The results show that increased vaccine acceptance guarantees recovery as newly reported cases dwindle. An exciting result indicates that the vaccine doses administered in China are tremendously effective in controlling the COVID-19 virus infections. As a result, death and newly confirmed cases will decline drastically.

## The importance of the COVID-19 vaccination on the tourism industry in China

China is the first country in Asia to develop and administer Sinopharm company's coronavirus vaccines, marking a significant leap forward in the international effort to control the virus and assist low- and middle-income countries in coping with the pandemic [73]. China's incredible effort and confidence in taking bold steps and initiatives have yielded significant results. It currently produces its vaccine and export it to about 28 countries. This confidence and positivity have further increased the hope of the tourism industry. Fig 10 demonstrates the trend of vaccination doses administered in China from 2020 through 2021. Thus, from December 15, 2020, to March 28, 2021, the number of COVID-19 vaccination doses administered in China has risen by millions.

As of March 28, 2021, China had administered approximately 107 million doses of the COVID-19 vaccine, compared to around 552 million globally. The vaccine inoculation

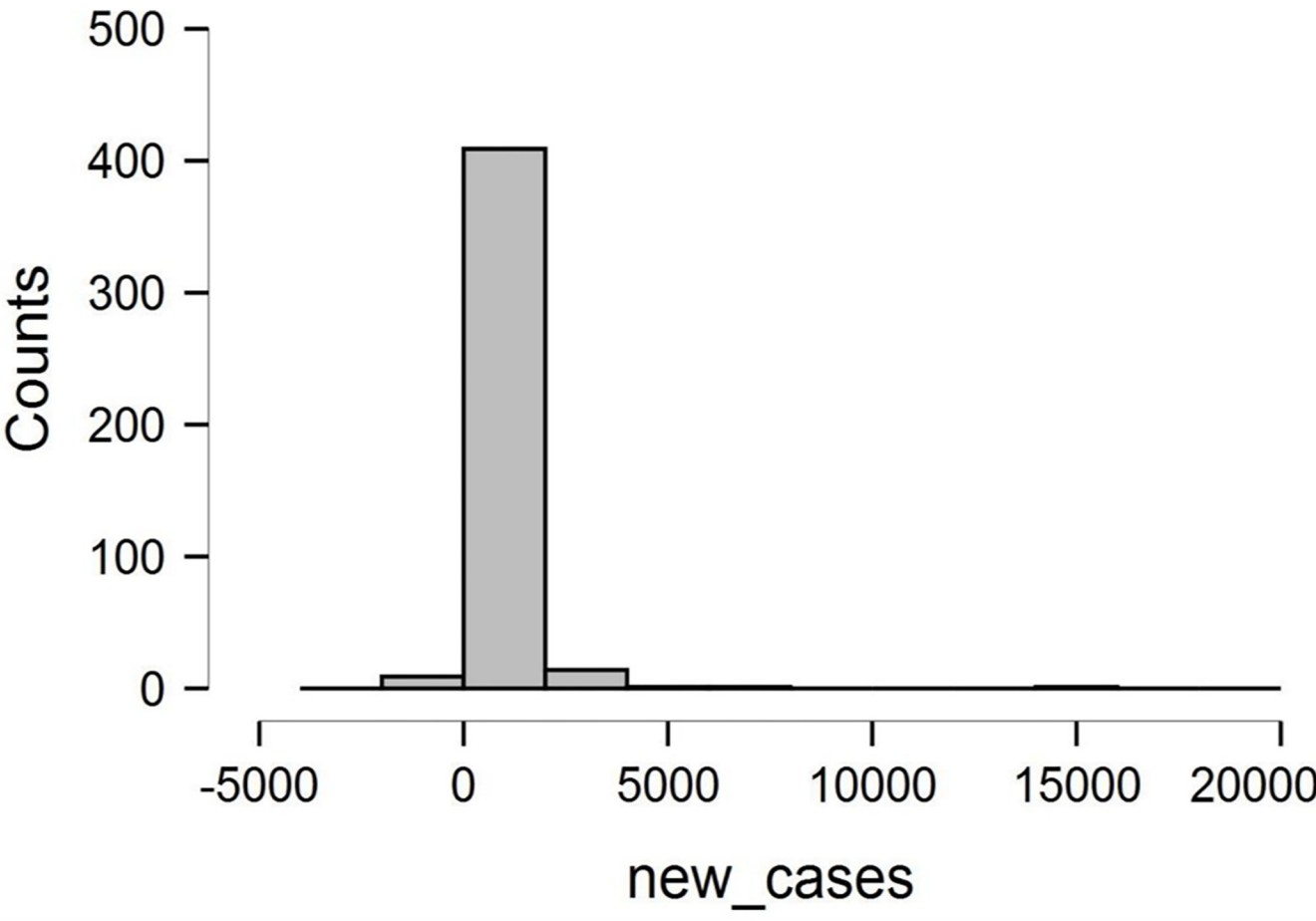

**Fig 6. Trends of new cases. Source:** Authors estimation (JASP Software).

acceptance numbers keep soaring as citizens have wholeheartedly accepted. According to current statistics, as of May 23, 2021, China had administered approximately 511 million doses of the vaccine, while about 1.67 billion doses had been applied globally. Furthermore, vaccines developed by Sinovac and Sinopharm pharmaceutical companies have received approvals and advanced purchase agreements in Brazil, Turkey, and Indonesia [75]. Figs 8 and 9 affirm that fewer recorded infection and death cases increase as the vaccine doses are administered.

In summary, the higher the vaccine acceptance signifies, the faster recovery from the pandemic, and the tourism industry can quickly revamp its losses. The result gives assurance that many visitors can travel to and from China to tour attractive sites, which will boost the performance of the tourist industry.

## Conclusion

This study emphasized the relevance of the COVID-19 vaccine on the tourism industry and the need for citizens' endorsement of the vaccine to ensure total recovery from the pandemic. Studies reviewed indicated that the unavailability of an effective vaccine for the treatment of infected persons was the most significant challenge confronting the management of the COVID-19 pandemic. This study found that the inception of the vaccines relieves the psychological and emotional stress of many citizens. Also, the study findings indicated that industries in tourism are hopeful of regaining the past losses. The study results further showed an

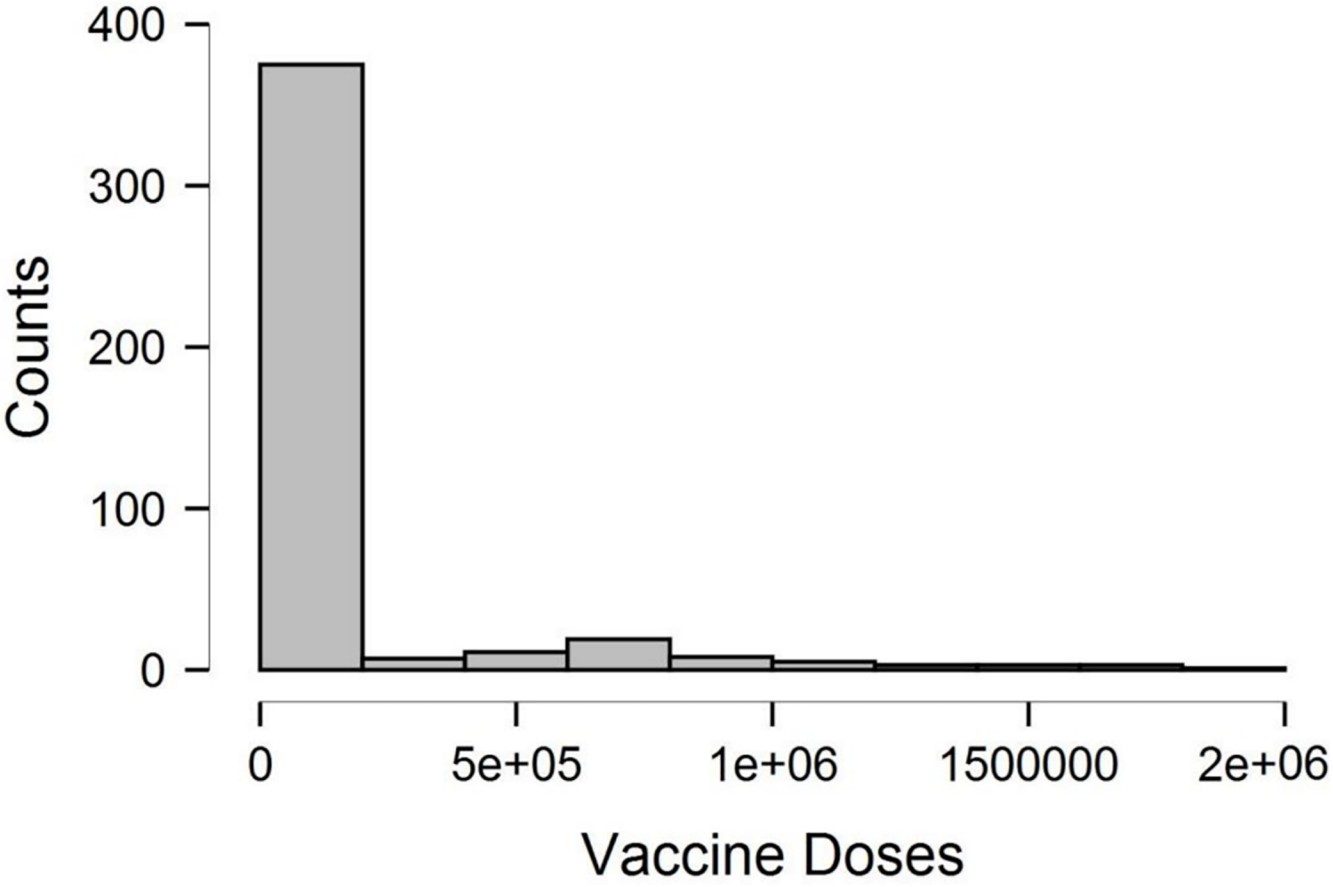

**Fig 7. Trend of vaccine doses. Source:** Authors estimation (JASP Software).

enormous decline in death and new cases. There was an increasing rate of vaccine acceptance by individuals, giving hope to the tourism industries and the entire businesses in the world.

The study findings showed that all the top 6 destinations involved in Australia's outbound tourism, including China, Indonesia, New Zealand, Thailand, the United Kingdom, and the United States, experienced a decline in their tourism activities and revenues due to the COVID-19 pandemic within 2020.

## Policy recommendations

As the COVID-19 virus infections continue to spread globally, healthcare workers and the government have admonished all citizens to accept the vaccine. Due to the recently Delta variant reported cases and infections, those who have not yet taken the vaccine may be at risk. In achieving faster results, the policymakers' should focus on assisting foreign nationals with easy access to vaccines. Based on the above reasons, this study suggests the following recommendation to the tourism stakeholders:

- Vaccination certificates of all visitors should be strictly checked. Further, hoteliers should put adequate measures to monitor all visitors who visit the various scenery sites.

- Through a collaborative effort between the tourism industry and the government, the following measures may be implemented to ensure effective control of the COVID-19 pandemic:

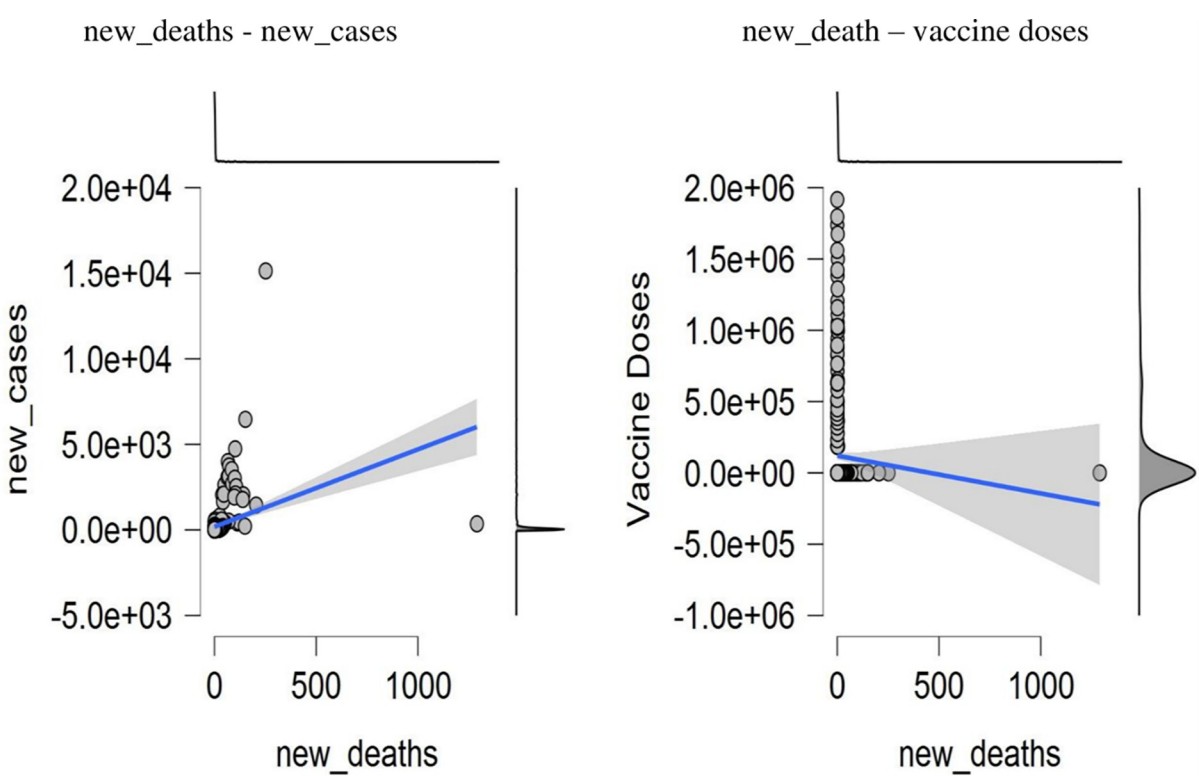

**Fig 8. Trend of new_deaths—new_cases and new_death–vaccine doses. Source:** Authors estimation (JASP Software).

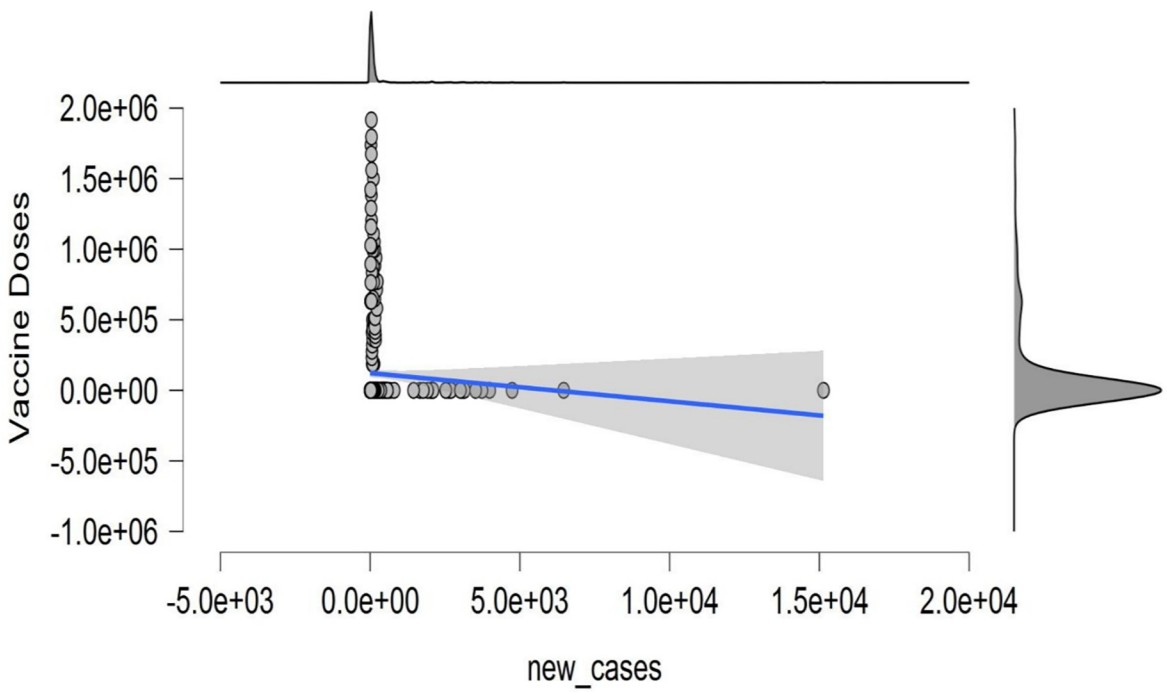

**Fig 9. Trend of new cases–vaccine doses. Source:** Authors estimation (JASP Software).

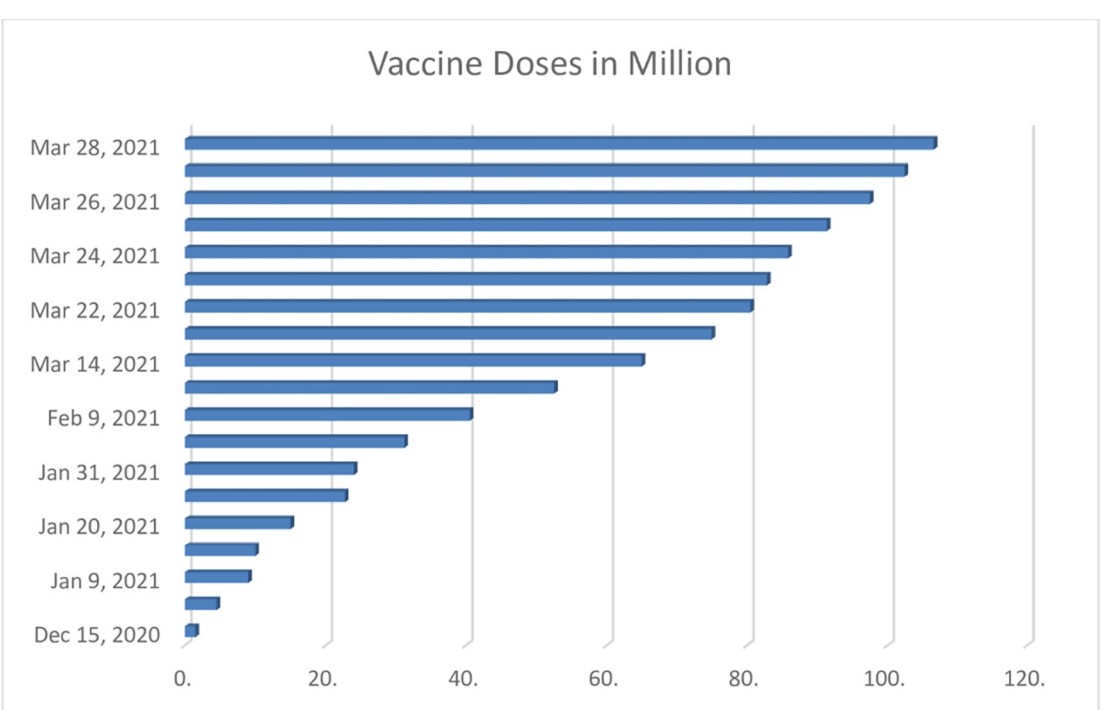

**Fig 10. Trends of COVID-19 vaccination doses in China. Source:** [74] modified by the researchers.

Firstly, all citizens must support the authorities' policies and structures outlined to control and prevent COVID-19. Healthcare professionals must be assigned to treat patients, trace contacts, conduct COVID-19 tests, and disinfect key areas within provinces and cities.

Secondly, all citizens are admonished to get vaccinated. Widespread vaccinations are a critical tool to achieve herd immunity. Exceptional circumstances of not taking the vaccine must be based on medical or health conditions backed by the healthcare provider's assessment of the inherent risks.

Again, it is recommended that individuals follow the latest updates from official sources. It is recommended that all inhabitants follow public accounts such as WeChat in China for current information in their respective provinces or cities. Updates from Foreign Affairs Offices and Health Commissions must be deemed official and authentic.

Finally, citizens are advised to keep themselves safe by following simple preventive measures such as physical distancing, wearing masks, keeping rooms aired, and avoiding crowds. It is believed that these are yet the most simple but effective and efficient ways to curb the spread of the Nobel coronavirus.

## Supporting information

**S1 File.**
(DOCX)

## Acknowledgments

The authors of this manuscript acknowledge the enormous contribution rendered by the National Natural Science Foundation of China.

## Author Contributions

**Conceptualization:** Fredrick Oteng Agyeman.

**Data curation:** Malcom Frimpong Dapaah.

**Formal analysis:** Mingxing Li.

**Methodology:** Zhiqiang Ma.

**Software:** Israel Adikah.

**Supervision:** Zhiqiang Ma.

**Writing – original draft:** Fredrick Oteng Agyeman.

**Writing – review & editing:** Agyemang Kwasi Sampene.

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
