## [Decision Letter · Decision Letter 0]

9 Feb 2022

PONE-D-21-34514The Relevance of the COVID-19 vaccine on the Tourism Industry: Evidence from ChinaPLOS ONE

Dear Dr. Agyeman,

Thank you for submitting your manuscript to PLOS ONE. After careful consideration, we feel that it has merit but does not fully meet PLOS ONE’s publication criteria as it currently stands. Therefore, we invite you to submit a revised version of the manuscript that addresses the points raised during the review process.

We look forward to receiving your revised manuscript.

Kind regards,

Helen Onyeaka, PhD

Academic Editor

PLOS ONE

Journal Requirements:

Reviewers' comments:

Reviewer's Responses to Questions

**Comments to the Author**

1. Is the manuscript technically sound, and do the data support the conclusions?

Reviewer #1: Yes

Reviewer #2: Yes

2. Has the statistical analysis been performed appropriately and rigorously? 

Reviewer #1: No

Reviewer #2: Yes

3. Have the authors made all data underlying the findings in their manuscript fully available?

Reviewer #1: Yes

Reviewer #2: Yes

4. Is the manuscript presented in an intelligible fashion and written in standard English?

Reviewer #1: Yes

Reviewer #2: No

5. Review Comments to the Author

Reviewer #1: I find the topic "The Relevance of the COVID-19 vaccine on the Tourism Industry: Evidence from China" current and relevant. The authors seek to establish the relevance of COVID-19 vaccine and the impact of the pandemic on China’s tourism. The authors employ a specification that uses vaccine doses as the dependent variable and the death and recorded new cases as independent variables.

Comments:

1. The authors employ a linear regression model. It is easy to check the underlying assumptions of the classical linear regression models. For instance one can plot the residuals against the fitted values in order to ascertain if heteroskedasticity is present. I recommend the authors to prove why a linear model was a parsimonious choice for this study.

2. Residual normality can also be checked via quantile-quantile plot or an appropriate statistical test.

3. It will also improve the paper if the authors can compare the predicted values of the model against the actual response variable (in a plot) to see if the model approximates the actual data.

In summary, I am missing some residual diagnostics analysis using appropriate statistical tests / plots which provides an indication about the performance of your model. Otherwise the results cannot be trusted.

4. Although well structured, the paper needs 'English Editing'. For example the opening sentence in the introduction needs to be reconstructed. In some cases authors wrote "independence variables" instead of independent variables" etc. In the introduction the authors also wrote that 'The availability of vaccines calms down citizens' nerves and improves

the citizens' psychological and mental health', such assertions are either tested and proved or cite existing studies to such assertions.

Reviewer #2: The title should read: Relevance of COVID-19 vaccine on the Tourism Industry:

Evidence from China

The Key words should read: COVID-19 vaccine, Tourism Industry, infection, new

cases; new death, China

Methodology Section

This section seems interesting as the flow of its introduction rolls out well.

Results and Discussion

Virtually all your tables lack appropriate specific titles.

For example, Table 1: Descriptive Statistics on what?

Table 3: ANOVA of what;

Table 4: Coefficients of what?

Still on the Results and Discussion section, the subsections/subtopics are not properly captured. For instance, “the pattern of Vaccine doses and impact on Death cases and New reported cases in China’ need to be recast.

More so, there is poor coherence between your results and discussion section

More so, majority of the tables conventionally lacks a source at the foot of the table

Can the figures mentioned in the results and discussion section be restored from beyond the references section?

Note that the first column in Table 1 should have a title which is suggested to be Measure

Your references are not bad, but there is room to recheck if each of them aligns with the accepted referencing style. More so, the researchers should ensure that all the citations are captured in the reference section.

The manuscript paper is accepted with major corrections.

6. PLOS authors have the option to publish the peer review history of their article (what does this mean?). If published, this will include your full peer review and any attached files.

Reviewer #1: No

Reviewer #2: No

---

## [Author Response · Author response to Decision Letter 0]

25 Mar 2022

Authors' Responses to the Editor and Reviewers' Comments

Submission of Revised Paper: The Relevance of the COVID-19 vaccine on the Tourism Industry: Evidence from China (PONE-D-21-34514). 

To begin with, the authors would like to express their profound gratitude to the Editors and the Reviewers for their valuable comments and suggestions raised to improve the manuscript. 

The responses to the Editors' and Reviewers' suggestions are given as follows.

Editor's comments 

Thank you for submitting your manuscript to PLOS ONE. After careful consideration, we feel that it has merit but does not fully meet PLOS ONE's publication criteria as it currently stands. Therefore, we invite you to submit a revised version of the manuscript that addresses the points raised during the review process.

Authors' Responses 

Thank you very much for your valuable information. Please, the authors have carefully revised the manuscript to meet PLOS ONE's publication criteria fully. The authors have answered and incorporated all the valuable suggestions and comments raised by the Editor and the Reviewers within the current version of the manuscript. Also, the authors have further conducted a round of proofreading of the entire manuscript. The authors are very grateful for your acceptance of the current version of the manuscript for publication in your highly cherished journal. 

Authors' responses to Reviewer's comments: 

Reviewer 1

Comments (1) I find the topic "The Relevance of the COVID-19 vaccine on the Tourism Industry: Evidence from China" current and relevant. The authors seek to establish the relevance of COVID-19 vaccine and the impact of the pandemic on China's tourism. The authors employ a specification that uses vaccine doses as the dependent variable and the death and recorded new cases as independent variables.

Authors' Response: Thank you very much for your commendation. The authors are very grateful.

Comments (2) The authors employ a linear regression model. It is easy to check the underlying assumptions of the classical linear regression models. For instance, one can plot the residuals against the fitted values in order to ascertain if heteroskedasticity is present. I recommend the authors to prove why a linear model was a parsimonious choice for this study.

Authors' Response: Thank you very much for your needful comments. The authors chose the linear regression model because of its widely recognized scientific methods that are reliable for analyzing and predicting data. The properties of linear regressions facilitate easy understanding and replicability. Further, the authors selected the linear regression model for this study analysis because of its significant explanatory predictive power. The Linear regression model helped explain the study data with a minimum number of parameters or predictor variables. Also, the result of this study indicated no heteroskedasticity or outliers. 

Comments (3): Residual normality can also be checked via quantile-quantile plot or an appropriate statistical test.

Authors' Response: The authors are very grateful for your valuable comments to improve the manuscript. We highly agree with your constructive comments and have incorporated the quantile-quantile plot within the current version of the manuscript. We have added the Q-Q plots as Supplementary material S3 Fig and S4 Fig. Also, this study’s analysis was formally conducted using the Shapiro-Wilk test, which helped check the data's normality assumptions. Thank you 

Comments (4): It will also improve the paper if the authors can compare the predicted values of the model against the actual response variable (in a plot) to see if the model approximates the actual data. In summary, I am missing some residual diagnostics analysis using appropriate statistical tests / plots which provides an indication about the performance of your model. Otherwise, the results cannot be trusted. 

Authors' Response: The authors are very grateful for your constructive comments. The authors have incorporated the analysis of the residual statistics and collinearity diagnostic within the current version of the manuscript Tables 5 and 6, respectively. Further supplementary material S2 Fig Residual versus covariate plot has been included in the current version of the manuscript. 

Comments (5): Although well structured, the paper needs 'English Editing'. For example, the opening sentence in the introduction needs to be reconstructed. In some cases, authors wrote "independence variables" instead of independent variables" etc. In the introduction the authors also wrote that 'The availability of vaccines calms down citizens' nerves and improves the citizens' psychological and mental health', such assertions are either tested and proved or cite existing studies to such assertions. 

Authors' Response: The authors are very grateful for your needful comments to improve the manuscript. Please the authors have made round proofreading of the entire manuscript. Thank you 

Reviewer 2

Comments (1): The title should read: Relevance of COVID-19 vaccine on the Tourism Industry: Evidence from China

Authors' Response: Thank you very much for your constructive comments. The authors have incorporated your valuable suggestions within the manuscript. 

Comments (2): The Key words should read: COVID-19 vaccine, Tourism Industry, infection, new cases; new death, China

Authors' Response: Thank you very much for your constructive suggestion to improve the manuscript. The authors have incorporated your valuable suggestions.

Comments (3): Methodology Section. This section seems interesting as the flow of its introduction rolls out well.

Authors' Response: Thank you very much for your commendation. The authors are very grateful.

Comments (4): Results and Discussion. Virtually all your tables lack appropriate specific titles. For example, 

Table 1: Descriptive Statistics on what?

Table 3: ANOVA of what;

Table 4: Coefficients of what? 

Authors' Response: Thank you very much for your constructive comments. The authors have made modifications to the table titles accordingly within the manuscript.

Comment (5) Still on the Results and Discussion section, the subsections/subtopics are not properly captured. For instance, “the pattern of Vaccine doses and impact on Death cases and New reported cases in China’ need to be recast.

Authors’ Response: Thank you very much for your needful comments to improve the manuscript's content. The authors have incorporated your valuable comments within the current version of the manuscript.

Comment (6) More so, there is poor coherence between your results and discussion section 

Authors’ Response: Thank you very much for your constructive suggestion to improve the manuscript content. The authors have restructured the results and discussion. 

Comments (7): More so, majority of the tables conventionally lacks a source at the foot of the table

Authors’ Response: The authors are very grateful for your needful comments and have incorporated your valuable suggestions. The authors generated all the output from the JASP software and have respectively provided footnotes for the tables required. Also, S1 Table, Australia’s total international tourism outbound from 2008 – 2020, has been provided with a reference source. 

Comments (8): Can the figures mentioned in the results and discussion section be restored from beyond the references section?

Authors’ Response: Thank you very much for your needful question. The authors have restored all the figures' captions within the manuscript. Also, all the figures have been prepared based on the journal's formatting style (using PACE) and have been included in the figure file accordingly. 

Comments (9): Note that the first column in Table 1 should have a title which is suggested to be Measure

Authors’ Response: Thank you very much. The authors appreciate your valuable suggestions to improve the manuscript. Please, we have incorporated your needful comments within the manuscript. 

Comments (10): Your references are not bad, but there is room to recheck if each of them aligns with the accepted referencing style. More so, the researchers should ensure that all the citations are captured in the reference section.

Authors’ Response: The authors are very grateful for your constructive comments to improve the manuscript. The authors have rechecked and aligned all the references to capture all the citations in the references section according to PLOS ONE Journal referencing style. 

Comments (11): The manuscript paper is accepted with major corrections.

Authors’ Response: Thank you for your valuable recommendation to accept our manuscript upon major corrections. The authors have comprehensively answered all questions and implemented the comments and suggestions made by the Editor and the reviewers. Therefore, we are hopeful that this current version is acceptable for publication and satisfies PLOS ONE publication criteria. 

Thank you very much

---

## [Decision Letter · Decision Letter 1]

17 May 2022

Relevance of COVID-19 vaccine on the tourism industry: Evidence from China

PONE-D-21-34514R1

Dear Dr. Oteng Agyeman,

We’re pleased to inform you that your manuscript has been judged scientifically suitable for publication and will be formally accepted for publication once it meets all outstanding technical requirements.

Kind regards,

Helen Onyeaka, PhD

Academic Editor

PLOS ONE

Additional Editor Comments (optional):

Reviewers' comments:

Reviewer's Responses to Questions

**Comments to the Author**

1. If the authors have adequately addressed your comments raised in a previous round of review and you feel that this manuscript is now acceptable for publication, you may indicate that here to bypass the “Comments to the Author” section, enter your conflict of interest statement in the “Confidential to Editor” section, and submit your "Accept" recommendation.

Reviewer #1: All comments have been addressed

2. Is the manuscript technically sound, and do the data support the conclusions?

Reviewer #1: Yes

3. Has the statistical analysis been performed appropriately and rigorously? 

Reviewer #1: Yes

4. Have the authors made all data underlying the findings in their manuscript fully available?

Reviewer #1: Yes

5. Is the manuscript presented in an intelligible fashion and written in standard English?

Reviewer #1: Yes

6. Review Comments to the Author

Reviewer #1: The authors addressed all the comments raised regarding the missing diagnostics such as plot of residuals. The authors do this by adding supplementary graphs to the paper.

7. PLOS authors have the option to publish the peer review history of their article (what does this mean?). If published, this will include your full peer review and any attached files.

Reviewer #1: **Yes: **Dr Phemelo Tamasiga

---

## [Editor Report · Acceptance letter]

10 Aug 2022

PONE-D-21-34514R1 

Relevance of COVID-19 vaccine on the tourism industry: Evidence from China 

Dear Dr. Oteng Agyeman:

I'm pleased to inform you that your manuscript has been deemed suitable for publication in PLOS ONE. Congratulations! Your manuscript is now with our production department. 

Kind regards, 

on behalf of

Dr. Helen Onyeaka 

Academic Editor

PLOS ONE